# Use of machine learning methods to understand discussions of female genital mutilation/cutting on social media

Gray Babbs[1☯¤a], Sarah E. Weber[1☯¤b]*, Salma M. Abdalla[1], Nina Cesare[2‡], Elaine O. Nsoesie[3‡]

**1** Department of Epidemiology, Boston University School of Public Health, Boston, Massachusetts, United States of America, **2** Biostatistics and Epidemiology Data Analytics Center, Boston University School of Public Health, Boston, Massachusetts, United States of America, **3** Department of Global Health, Boston University School of Public Health, Boston, Massachusetts, United States of America

☯ These authors contributed equally to this work.
¤a Current address: Department of Health Services, Policy and Practice, Brown University School of Public Health, Providence, Rhode Island, United States of America
¤b Current address: Section of Infectious Diseases, Boston Medical Center, Boston, Massachusetts, United States of America
‡ NC and EON also contributed equally to this work.
* sarah.weber@bmc.org

**Data Availability Statement:** We provide datasets containing tweet IDs for each of the six years in the dataset in a repository available here: https://github.com/graybabbs/fgmctwitter/ Original tweet

## Abstract

Female genital mutilation/cutting (FGM/C) describes several procedures that involve injury to the vulva or vagina for nontherapeutic reasons. Though at least 200 million women and girls living in 30 countries have undergone FGM/C, there is a paucity of studies focused on public perception of FGM/C. We used machine learning methods to characterize discussion of FGM/C on Twitter in English from 2015 to 2020. Twitter has emerged in recent years as a source for seeking and sharing health information and misinformation. We extracted text metadata from user profiles to characterize the individuals and locations involved in conversations about FGM/C. We extracted major discussion themes from posts using correlated topic modeling. Finally, we extracted features from posts and applied random forest models to predict user engagement. The volume of tweets addressing FGM/C remained fairly stable across years. Conversation was mostly concentrated among the United States and United Kingdom through 2017, but shifted to Nigeria and Kenya in 2020. Some of the discussion topics associated with FGM/C across years included Islam, International Day of Zero Tolerance, current news stories, education, activism, male circumcision, human rights, and feminism. Tweet length and follower count were consistently strong predictors of engagement. Our findings suggest that (1) discussion about FGM/C has not evolved significantly over time, (2) the majority of the conversation about FGM/C on English-speaking Twitter is advocating for an end to the practice, (3) supporters of Donald Trump make up a substantial voice in the conversation about FGM/C, and (4) understanding the nuances in how people across cultures refer to and discuss FGM/C could be important for the design of public health communication and intervention.

text and user metadata reported in this study cannot be deposited in a public repository because of Twitter's data sharing policy. Tweet IDs can be used to retrieve original tweet data via Twitter's public application programming interface (API).

**Funding:** The authors received no specific funding for this work.

**Competing interests:** The authors have declared that no competing interests exist.

## Introduction

Female genital mutilation/cutting (FGM/C) is an umbrella term for procedures that involve injury to the vulva and vagina for nontherapeutic reasons [1]. At least 200 million women and girls living in 30 countries have undergone FGM/C [2]. Rates of FGM/C are generally decreasing, but the rates by which they are decreasing vary by country and region [3, 4]. FGM/C is most prevalent in Eastern, Northeastern, and Western Africa and also occurs among migrants and refugees worldwide [1]. FGM/C is practiced across religions and is not solely associated with any religious group [5, 6]. FGM/C is illegal in many countries, including nearly all of the countries where it is most commonly practiced [7].

FGM/C is considered a violation of human rights law and a pressing public health issue [8]. It is associated with a number of negative short-term health outcomes, including hemorrhage, shock, and infection, as well as long-term physical complications, including chronic infections, painful urination, sexual health challenges, and obstetric complications [9]. Studies have also pointed to a relationship between FGM/C and adverse mental health outcomes, including post-traumatic stress disorder (PTSD), anxiety and depression [10].

Negative public discourse on FGM/C can perpetuate stigma for women and girls in diasporic communities that practice (or are perceived to practice) FGM/C [11]. A qualitative study reported that people who have undergone FGM/C feel pressure to lie about their FGM/C status to avoid stigmatization or social exclusion in their new cultural contexts [12]. Other studies note that women who have undergone FGM/C in the diaspora report a perceived sense of "feeling different" from their peers [13, 14], above and beyond differences due to immigration status and race/ethnicity alone [15]. This stigma causes many women to completely avoid discussion about FGM/C [14].

Analysis of social media discussion offers the opportunity to assess sentiments, knowledge and attitudes towards FGM/C across the globe, without requiring the resources needed for a large-scale survey. Social media platforms' perceived anonymity can bring traditionally private conversations into the public sphere, serving as valuable resources for understanding perceptions of stigmatized issues [16–18]. Social media data are uniquely positioned to track public conversations over time, particularly for sensitive topics that may be difficult to capture via surveys such as miscarriage, preterm birth [19] and circumcision [20]. Moreover, social media offers the opportunity to analyze how issues are framed within a conversation space, as they contain longitudinal information and user-to-user replies. However, few studies to date have used social media data to analyze FGM/C discussions [21].

In this paper, we use social media data to assess public perception of FGM/C on Twitter between 2015 and 2020. First, we characterize general discussions of FGM/C on English-speaking Twitter. Next, we analyze trends in discussion of FGM/C discussions on Twitter over the study period. Lastly, we characterize the meta-data of the discussion—what terms are people using to describe FGM/C, who is engaging in the conversation, and what influences engagement. Our findings can add to the current literature about public perception about FGM/C. Further, our work can be used by medical professionals, public health practitioners, governmental agencies, and advocacy organizations seeking to educate the public about FGM/C.

## Materials and methods

### Data extraction

We extracted tweets via Twitter's (twitter.com) Application Programming Interface (API), which provided a random 1% sample of all public tweets containing specific keywords. Tweets

in our corpus contained at least one of the following terms: "FGM," female genital mutilation," "female genitalia mutilation," "female circumcision," "female genital alteration," "female genitalia alteration," "female genital cutting," "female genitalia cutting," "female genital excision," and "female genitalia excision." Tweets were posted between January 1, 2015 and December 31, 2020. We restricted our analysis to tweets in English. The research was conducted in line with Twitter API's requirements for academic research [22].

## User characteristics

To understand user volume, we compared the number of users who tweeted about FGM/C each year. We also looked at the number of users with verified accounts, that is the number of accounts that are designated by Twitter as "active, notable, and authentic accounts of public interest". We also looked at the proportion of users each year who were newly engaged, defined as users who tweeted about FGM/C for the first time for each year during the study period.

We examined top terms found in user descriptions by normalizing, stemming, and removing all stop words from user description text and calculating the frequency of each word used. We extracted the top 15 words used each year. Overall, a total of 25 words were included in at least one year's top 15 most frequently used words. Finally, we calculated the proportion of user descriptions each year containing each word and compared over time.

Using extracted country names, we calculated the proportion of total users each year who identified each country in their user location, and evaluated if the countries people are most frequently tweeting from changed across years.

## Characterizing FGM/C conversations over time

We measured the number of tweets, retweets and likes in each year of the study period. We also measured tweet volume on International Day of Zero Tolerance for FGM by assessing number of tweets on February 6th of each year in the study period.

We also evaluated the use of FGM/C terminology by determining the frequencies of the terms "female circumcision," "cutting," "mutilation," "excision," and "alteration." Presence of terms in a tweet were not mutually exclusive.

Top words in tweet text were found using the same method as the user description. We extracted the top 15 words used each year and calculated the proportion of tweets each year containing each word. In total, 34 words were included, as this was the number of words in the top 15 word lists after deduplicating those that repeated over years.

To characterize conversations about FGM/C, we used Correlated Topic Modelling (CTM) to identify major topics in our corpus [23, 24]. CTM is an extension of latent Dirichlet allocation (LDA) topic modeling, which assumes that documents (tweets) are generated from a finite group of topics. CTM is distinguished, however, by the recognition that topics within a document may be correlated with one another. We ran CTM models separately for each year of analysis. Topics were inferred by the research team based on groupings of terms that were most commonly found in and most unique to a topic. Topics were cross-validated by referencing the text of tweets containing each topic's unique terms. Ten topics were selected for each year. After exploratory analysis, our team decided that ten topics balanced maximizing identifiable constructs in the data and minimizing redundancy across topics. Reported example tweets have been modified slightly to preserve user anonymity, according to best practices [25]. Tweet topics that were specific to an event or particular year were synthesized using only tweets from the year the topic was identified.

## Understanding user engagement

We analyzed tweet and user characteristics associated with greater user engagement using two classifiers–support vector machine (SVM) and random forest models, which have been used in similar studies [26, 27]. We fit models first on all of the years in the data. We then fit models on each year's subset, using the number of retweets to represent user engagement. Number of retweets was classified into two categories: high retweets were defined as the year's 90th retweet percentile or above, and low retweets were a random matched sample of tweets with zero retweets. We chose retweet count as the representation of user engagement because it had little missingness compared to like count, favorite count, reply count, and quote count. We split all user engagement models into train and test sets using a 70/30 split ratio.

We optimized the random forest model with varying numbers of trees and inclusion of predictors. Account level variables in the model included number of accounts followed, number of accounts following, number of tweets tweeted, and account verification. Tweet level variables included tweeting on a high volume date (defined as a top 11 date for the dataset), tweet length, and inclusion of key FGM/C terminology in the tweet ("alteration," "circumcision," "cutting," "excision," "mutilation," "fgm"). We compared accuracy between the SVM and random forest models and ran separate models for each of the six study years. All analyses were completed in R Version 4.0.2.

## Results

### User characteristics

The final dataset contained 1,301,203 tweets about FGM/C. Of these tweets, 1,004,937 were original tweets and 296,266 were retweets. During the study period from 2015 to 2020, 338,845 unique users tweeted about FGM/C. The most users were engaged in conversations in 2017 and 2018, and the least users were engaged in 2016 (Table 1). The number of users decreased steadily from 2017 to 2020. Users were similar from 2015–2016; only 23.4% of users in 2016 had not previously tweeted about FGM/C in 2015. However, in 2017, 83.7% of users joined FGM/C discussions for the first time during the study period and the percentage of newly engaged users remained high from 2018–2020. A similar percentage of users each year were verified, excluding a spike in verified users in 2016.

Every year from 2015 to 2020, the most common terms in user descriptions were "views," "news," "health," and "love" (Fig 1). Other prevalent terms included "world," "writer," "social," "rights," "life," and "Trump." Notably, "MAGA" and "conservative" were also among the top user description terms. When comparing user description terms from year-to-year, there were no considerable changes in term prevalence over time (S1 Table).

We linked 100,267 (30%) of our 338,845 unique users to countries based on self-reported locations in user profiles. The conversation was concentrated among six countries (United

**Table 1. Number of users engaged in conversations around FGM/C over the study period and frequency of new and verified users within each year.**

| Year | Total Users | New Users N (%) | Verified Users N (%) |
|---|---|---|---|
| 2015 | 82,034 | — | 3,759 (4.6) |
| 2016 | 26,291 | 6,160 (23.4) | 1,840 (7.0) |
| 2017 | 95,162 | 79,669 (83.7) | 3,942 (4.1) |
| 2018 | 90,368 | 63,300 (70.0) | 3,622 (4.0) |
| 2019 | 86,978 | 57,711 (66.4) | 3,176 (3.7) |
| 2020 | 72,957 | 49,971 (68.5) | 2,483 (3.4) |

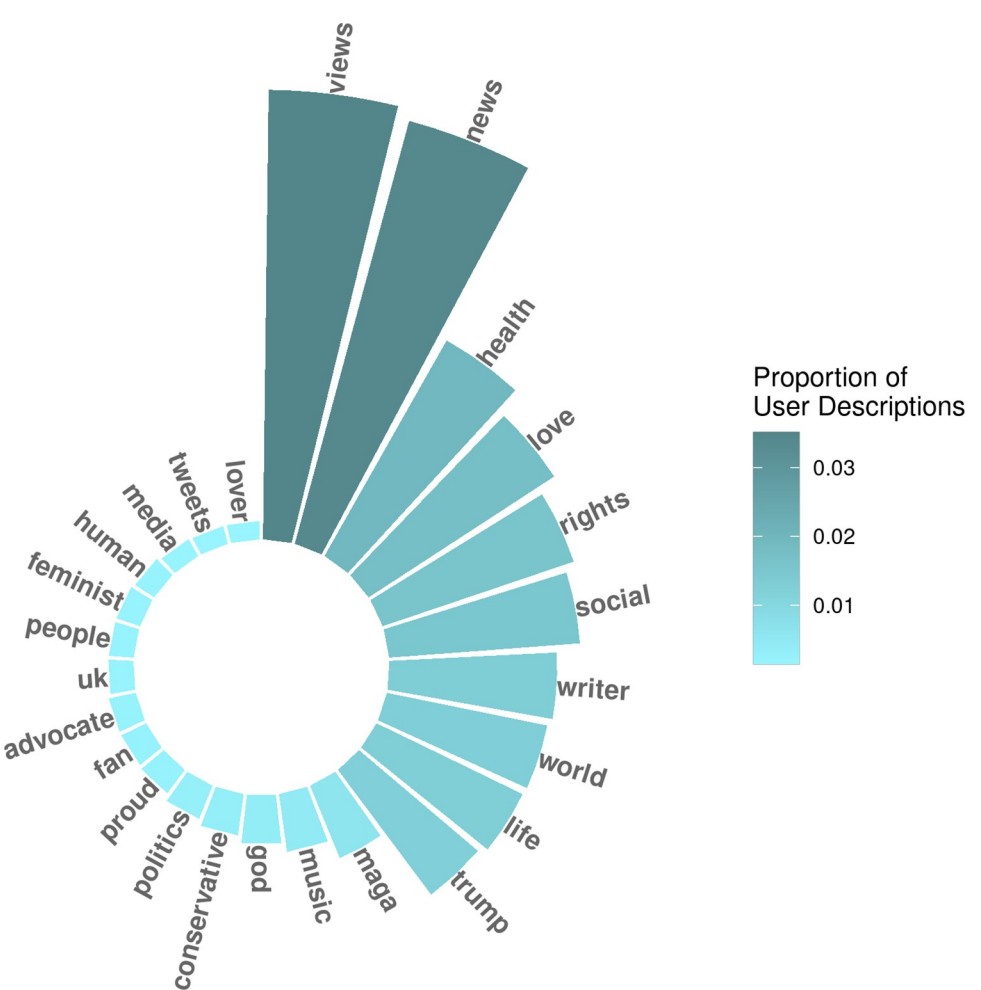

**Fig 1. Most prevalent terms in user descriptions in 2020.**

States, United Kingdom, Nigeria, Kenya, India and Australia), though relative proportions changed over time (Fig 2). In 2015 and 2016, the highest proportion of users were from the UK (20% and 21% respectively), followed by the United States, Nigeria, Kenya and Australia. From 2017 to 2019, users were heavily concentrated in the United States (between 19% and 24% of all users during this time). In 2020, the countries with the highest proportion of users were Kenya (15%) and Nigeria (14%), followed by the USA (10%), UK (10%), and India (7%).

## Characterizing FGM/C conversations over time

The overall volume of tweets addressing FGM/C remained moderately stable between 2015 and 2020 (Fig 3). We observed the greatest number of tweets (291,187) in 2017. This represents a two-fold jump in the number of tweets over the preceding year. Likes and retweets increased steadily between 2016 and 2018, after which likes continued to increase and retweets dropped.

In each of the study years, the highest volume of FGM/C specific tweets occurred on February 6, International Day of Zero Tolerance for Female Genital Mutilation, representing 4.8% of the entire dataset. The average daily tweet volume on this date was 17 times higher than the daily average.

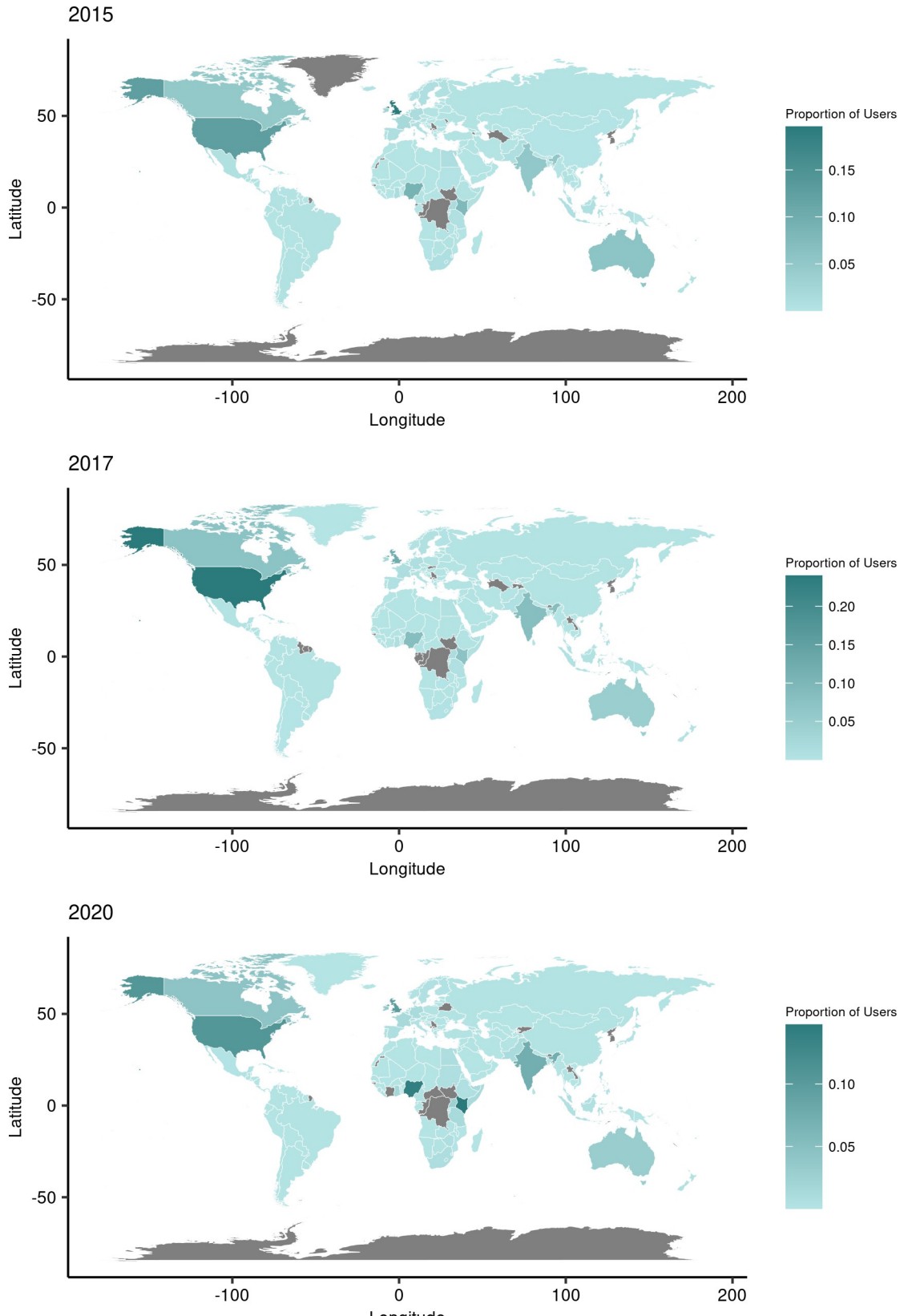

**Fig 2. User geographic distribution in 2015, 2017 and 2020.** Map base layer data sourced from Natural Earth, available from
https://www.naturalearthdata.com/.

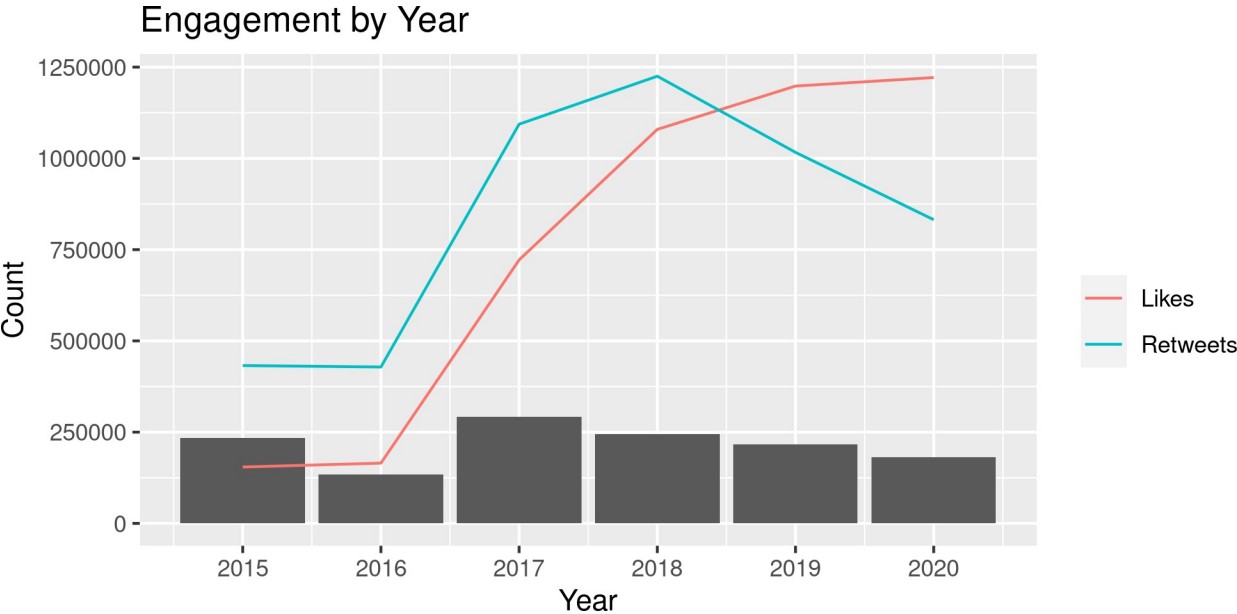

**Fig 3. Twitter engagement by year.** Tweet count is represented by gray bars, like count is represented by the blue line, and retweet count is represented by the pink line.

We explored the frequency of words used to describe FGM/C by identifying the words that we used to extract tweets. The terms that were used to refer to FGM/C remained fairly stable across the years studied. Compared to female genital mutilation or FGM, cutting (4.3%), female circumcision (2.2%), excision (0.1%), and alteration (<0.1%) were less commonly used terms for this practice. The words most commonly found in tweet text in 2020 included "endfgm," "uk," "egypt," and "end" (Fig 4). Other prevalent terms in tweet text from 2015–2020 included "circumcision," "child," and "news." Locations were often referenced, including the UK, Nigeria, Michigan, Kenya and Egypt. "Muslim," "Islam," and "Sharia" were also among the most commonly used words in the tweet text. The prevalence of common terms in tweet text did not change significantly over time (S2 Table).

Despite similarity in commonly referenced terms, topics identified using topic modeling varied across years (Table 2). Overall, Islam and the International Day of Zero Tolerance for Female Genital Mutilation were the most common topics identified. Both topics were present in all six years of analysis. Activism was the next most persistent topic, occurring in each year after 2015. Comparisons between FGM/C and male circumcision arose in 2017 and were present for every subsequent year. Conversations regarding the legality of FGM/C were identified in four years (2015, 2016, 2018, and 2019).

In 2017, 2018, 2019, and 2020, an anti-immigrant topic emerged that put FGM/C in a category with other violences against women ("acid attacks," "honor killing," "gang rape," and "child marriage"). Education around FGM/C was a topic present from 2015–2017, but it was not present from 2018–2020. This topic includes terms that promote FGM/C awareness such as "speak," "question" and "believe." FGM/C as a culturally significant practice was a topic identified across three years: 2015, 2016 and 2018.

Other themes identified across multiple years included medical risks (2015, 2019), human rights (2018, 2020), and feminism (2017, 2020). A number of topics could be linked to specific events, such as the implementation of stricter laws against FGM/C in Eritrea (2015) [28], The Gambia banning FGM/C (2015) [29], Egypt's parliament approving stringent penalties for

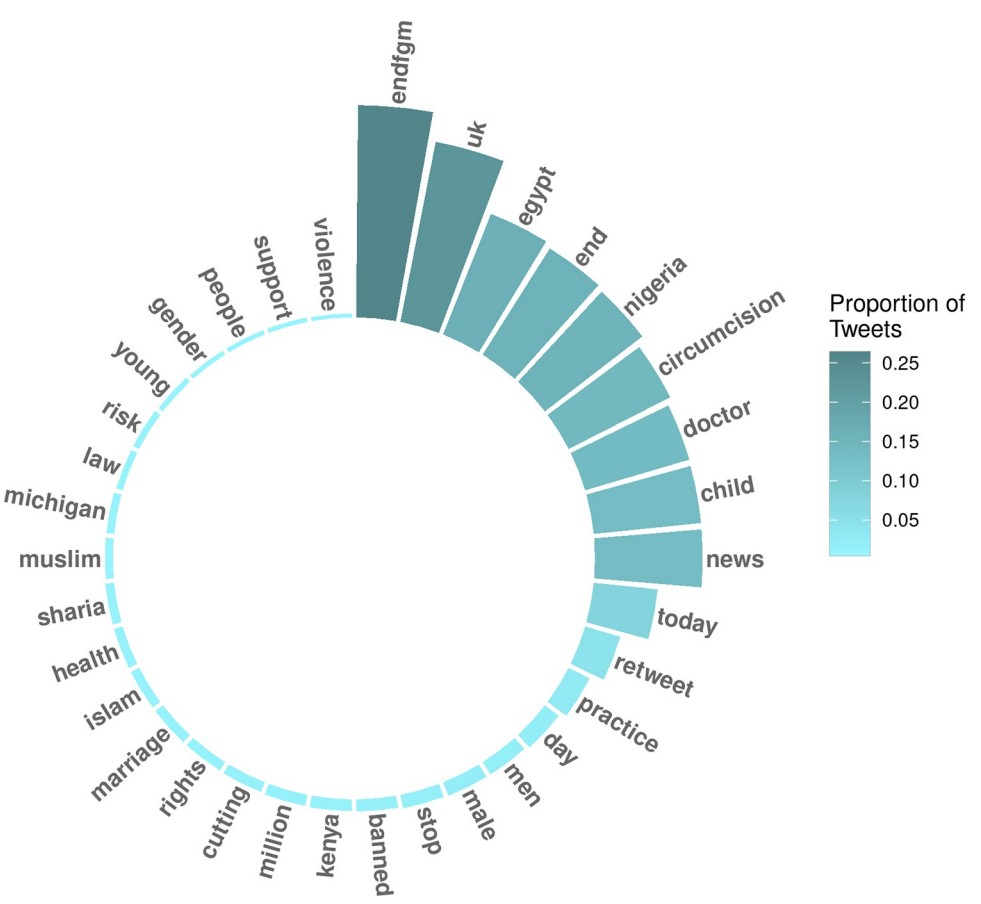

**Fig 4. Most prevalent terms used in Tweets in 2020.**

FGM/C (2016) [30], a doctor in Michigan, USA being charged with performing FGM/C (2017) [31, 32], and Sudan outlawing FGM/C (2020) [33, 34]. In 2020, COVID-19 emerged as a theme that was correlated with FGM/C discussions.

## User engagement

On average, original tweets received 3.86 retweets (standard deviation (SD) = 53.88) and all tweets received an average of 3.49 favorites (SD = 93.90). Using the full dataset, the random forest model performed better for predicting user engagement, with 86.3% accuracy (F1: 0.8621) compared to 73.0% in the SVM model (F1: 0.7440). Results from the random forest model suggest that factors predicting engagement (high volume date, tweet length, and inclusion of key FGM/C terminology in the tweet, number of accounts followed, number of accounts following, number of tweets tweeted, and account verification) were stable across time (Fig 5). Number of followers was consistently the first or second most important predictor, with the highest mean decrease in Gini score in 2015 and 2017 (S3 Table). Longer tweet length was the most important predictor in 2016 and 2018–2020. Variables such as the following count, number of total tweets, and whether the user had been verified, were also important predictors of whether a tweet would garner a high number of retweets. The presence of common words for FGM/C in the tweet text were of relatively little importance in the model. The model accuracy generally increased in later years.

**Table 2. Topics identified within FGM/C Tweets by year.**

| | 2015 | 2016 | 2017 | 2018 | 2019 | 2020 | Tweet Example |
|---|---|---|---|---|---|---|---|
| Islam | ▓ | ▓ | ▓ | | | | "I'm personally really ashamed that fgm is practiced in so much of the muslim world. It's so cruel. . ." |
| International Day of Zero Tolerance for FGM | ▓ | ▓ | ▓ | | | | "Today is International Day of Zero Tolerance. What is our government doing to stop FGM? #endFGM" |
| Current News Stories | ▓ | ▓ | ▓ | ▓ | | | "Michigan Doctor Arrested for Female Genital Mutilation" |
| Education | ▓ | ▓ | ▓ | ▓ | | | "Everyone needs to be educated on #FGM to protect vulnerable girls." |
| Legal | ▓ | ▓ | | ▓ | ▓ | | "Sierra Leone is close to outlawing FGM. Huge win." |
| Cultural Aspects | ▓ | ▓ | | ▓ | ▓ | | "FGM is a cultural practice- not a religious one. Culture can change. . ." |
| Medical Risks | ▓ | | | | | | "I saw the lacerations, the fistulas, the difficulties giving birth after. FGM is the most harmful thing that can happen to a girl." |
| Activism | | ▓ | ▓ | ▓ | ▓ | ▓ | "New challenges in the fight to ban FGM" |
| (Male) Circumcision | | | ▓ | ▓ | ▓ | ▓ | "Why does no one talk about male genital mutilation?" |
| Anti-immigrant Sentiments | | | ▓ | ▓ | | | "I don't consider FGM/honour killings/forced marriages /Sharia/Halal/terrorism etc. to be part of British culture. It's sad that you do." |
| Feminism | | | ▓ | ▓ | | | "FGM is why we need feminism" |
| Human Rights | | | | ▓ | ▓ | | "FGM isn't a female problem. It's a human rights problem." |
| Additional Identified Topics | | Violence; UNICEF | | | | | Violence: "Worldwide, 1 in 3 women experience sexual or physical violence. 200 million have undergone FGM. #endviolenceagainstwomen" <br> UNICEF: "Rates of FGM are decreasing in Africa but distubingly increasing in Indonesia." (Note: this topic corresponded with the 2016 release of a UNICEF report about FGM/C.) [2] |

A shaded box indicates that a topic was found in the topic model for a given year subset. Example tweets have been modified slightly to preserve user anonymity.

## Discussion

In this study, we aimed to characterize discussions of FGM/C on the social media platform, Twitter, from 2015 to 2020. In general, we found that discussion about FGM/C has not changed significantly over time. However, there were differences in the number of newly engaged users over the years, suggesting that users differed across years but users with similar views, backgrounds, and intentions may be drawn to Twitter as a platform for discussing these topics. While tweet count steadily decreased after 2017, retweet and like counts remained well above pre-2017 levels. This may suggest that although fewer original ideas about FGM/C were circulating post-2017, users were more familiar with the topic and more likely to retweet other voices. Major countries where conversation was concentrated remained consistent; however, the country with the highest volume of users shifted from the UK to the United States in 2017, until Nigeria and Kenya became the top countries in 2020.

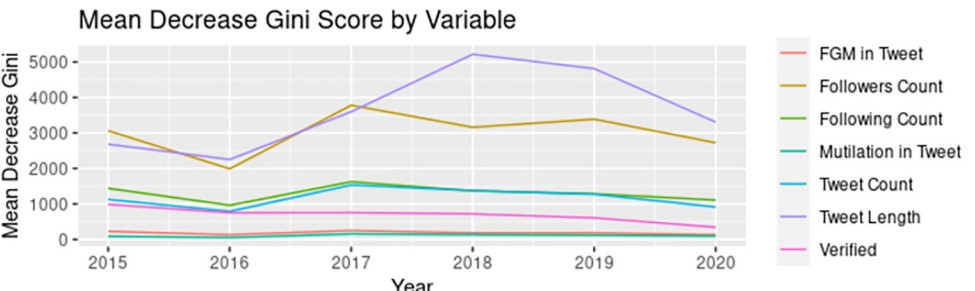

**Fig 5. Mean decrease Gini score by Tweet and user characteristics.**

We identify substantial increases in conversation related to five news stories in the study period: stricter laws against FGM/C in Eritrea (2015), The Gambia banning FGM/C (2015), Egypt's parliament approving stringent penalties for FGM/C (2016), a doctor in Michigan, USA charged with performing FGM/C (2017), and Sudan outlawing FGM/C (2020). Islam as a topic was identified across all years and "Muslim," "Islam," and "Sharia" were among the most common words in tweet text. These findings suggest that users often associated FGM/C with Islam, despite FGM/C occurring across religions.

Our results suggest that a substantial contingent of the conversation happening about FGM/C on Twitter (in English) is advocating for an end to FGM/C. The hashtag "endfgm" was the most common unique word identified in tweet text. Each year in the study period, we saw a substantial surge in tweets for International Day of Zero Tolerance for FGM, a 17-fold increase over the daily average. This suggests that observing specific days when FGM/C is recognized globally can increase awareness since individuals are likely to engage in conversation and advocacy on such days. Across the time period, we observe a shift from raising awareness about FGM/C (education, discussion of cultural aspects) to more explicit calls to an end of the practice. In the later portion of the study period, we see more movement-based language arise, tying FGM/C to feminism and struggles for human rights.

Another significant portion of the conversation used FGM/C as a rhetorical tool to associate FGM/C with Islam. A new topic arose in 2017 with anti-immigrant sentiments. It connected FGM/C to other practices such as child marriage, acid attacks, female infanticide, and gang rape. These tweets referenced that these practices were not part of British or American culture and made out countries where FGM/C is common to be backwards. This shift coincides with a 2017 increase in the proportion of users in the US engaging in FGM/C conversations. "MAGA, "Trump," and "conservative" were three of the most common words in Twitter user profiles of people tweeting about FGM/C, suggesting a strong voice from conservative Trump supporters in the FGM/C conversation. FGM/C sentiment that connects the practice closely with other violences against women that are relatively uncommon in the US and UK may further stigmatize people who have undergone FGM/C by painting "narratives of victimhood" [35].

These findings reveal known and novel insights about FGM/C-related public health communication and practice on Twitter. First, on English-speaking Twitter, the term FGM was most frequently used to describe the practice. Second, endfgm was the single most common word identified in our dataset. Use of this hashtag connected public health communication with the greater FGM/C advocacy conversation and emphasized supporting survivors and uplifting their voices. Third, given that user verification was the best predictor of engagement, verified accounts had a particularly large role to play in engaging the public in education about FGM/C. In addition, these results suggest that the conversation about FGM/C is increasingly politicized in English-speaking contexts, particularly in the UK and US, as seen by the strong conservative voice in the conversation; a fact well-known by researchers who study FGM/C [36].

One major limitation of this work is that it is restricted to conversations happening about FGM/C on Twitter. Twitter users are not representative of the global public, and we have limited information on demographic characteristics of users. Conversations offline or on different social media sites may vary from the findings here. Further, the data are not definitively representative of all conversations or users. Though tweets were collected from users around the world, we only analyze discourse among English language tweets. Tweets about FGM/C were identified based on presence of certain terms, so we may overlook conversations about FGM/C that do not use these terms, particularly terms for FGM/C in languages other than English. Lastly, topic models were run separately for each year of data. As such, the models are not directly comparable and may contain slightly different identified terms.

Despite these limitations, this study uses a large sample size to characterize conversations around this important public health topic. With over a million original tweets posted between 2015 and 2020, these data span a wide range of populations and conversations. It is the first study to our knowledge to evaluate FGM/C related social media discussions using machine learning methods. Though some surveys have been conducted about public perception about FGM/C, many of these surveys used direct questioning. Past studies have indicated that direct questioning about FGM/C, particularly among people who have been engaged in anti-FGM/C related programming, may lead to misreporting due to social desirability bias [37]. Social media platforms such as Twitter have substantial potential to study sensitive issues like FGM/C, as they do not employ direct questioning, but rather observe ongoing conversations. This kind of data acquisition poses unique challenges as well. Since the time of tweet extraction, Twitter has changed its policy around providing tweets to researchers twice [38].

Future studies should explore what proportion of social media conversations about FGM/C are from people immediately affected by the practice. This work may also address how discussion of FGM/C is impacted by events (e.g., media, news coverage, or celebrity posts). There is room to explore in greater qualitative detail how discourse varies by country, and whether negative discourse can be linked, on an aggregate level, to stress or negative health outcomes. Given the usage of such platforms for sharing health information and misinformation, it is important to understand how discussion about topics such as FGM/C are impacted by misinformation and political discourse. With additional research, public health practitioners, nongovernmental organizations, and governments can educate and design targeted interventions to reach communities and countries where the practice is ongoing.

These analyses add to a growing body of work that uses social media conversations to characterize stigmatized or misunderstood issues. Using Twitter data allows us to keep a finger on the pulse of public discourse to understand perceptions, develop public health communications, and hone more effective interventions [39].

## Supporting information

**S1 Table. Most common words found in user descriptions and the proportion of user descriptions containing each word by year.** Words are ordered based on their prevalence in 2015, followed by subsequent years.
(DOCX)

**S2 Table. Most common words found in Tweet text and the proportion of Tweets containing each word by year.** Words are ordered based on their prevalence in 2015, followed by subsequent years.
(DOCX)

**S3 Table. Ranked importance of Tweet and user characteristics in predicting high retweets in random forest model.** The importance is ranked ordinally. The factor listed as "1" is the first most influential factor in engagement, the factor listed as "2" is the second most important, and so on.
(DOCX)

## Author Contributions

**Conceptualization:** Elaine O. Nsoesie.

**Data curation:** Elaine O. Nsoesie.

**Formal analysis:** Gray Babbs, Sarah E. Weber, Nina Cesare, Elaine O. Nsoesie.

**Funding acquisition:** Elaine O. Nsoesie.

**Investigation:** Gray Babbs, Sarah E. Weber, Nina Cesare, Elaine O. Nsoesie.

**Methodology:** Gray Babbs, Sarah E. Weber, Nina Cesare, Elaine O. Nsoesie.

**Supervision:** Salma M. Abdalla, Nina Cesare, Elaine O. Nsoesie.

**Visualization:** Gray Babbs, Sarah E. Weber, Nina Cesare, Elaine O. Nsoesie.

**Writing – original draft:** Gray Babbs, Sarah E. Weber, Nina Cesare.

**Writing – review & editing:** Salma M. Abdalla, Nina Cesare, Elaine O. Nsoesie.

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
