## [Decision Letter · Decision Letter 0]

24 Jan 2023

PGPH-D-22-01078

Use of machine learning methods to understand discussions of female genital mutilation/cutting on social media

Dear authors,

Thank you for submitting your manuscript to PLOS Global Public Health. After careful consideration, we feel that it has merit but does not fully meet PLOS Global Public Health’s publication criteria as it currently stands. Therefore, we invite you to submit a revised version of the manuscript that addresses the points raised during the review process.

Please submit your revised manuscript by February 15th.  If you will need more time than this to complete your revisions, please reply to this message or contact the journal office at globalpubhealth@plos.org. Please include the following items when submitting your revised manuscript:

We look forward to receiving your revised manuscript.

Kind regards,

Maria del Mar Pastor Bravo, Ph.D.

Academic Editor

Journal Requirements:

1. In your Methods section, please include additional information about your dataset and ensure that you have included a statement specifying whether the collection and analysis method complied with the terms and conditions for the source of the data.

2. In the online submission form, you indicated that your data will be submitted to a repository upon acceptance.  We strongly recommend all authors deposit their data before acceptance, as the process can be lengthy and hold up publication timelines. Please note that, though access restrictions are acceptable now, your entire data will need to be made freely accessible if your manuscript is accepted for publication. This policy applies to all data except where public deposition would breach compliance with the protocol approved by your research ethics board. If you are unable to adhere to our open data policy, please kindly revise your statement to explain your reasoning and we will seek the editor's input on an exemption. Please be assured that, once you have provided your new statement, the assessment of your exemption will not hold up the peer review process.

3. Some material included in your submission may be copyrighted. According to PLOS’s copyright policy, authors who use figures or other material (e.g., graphics, clipart, maps) from another author or copyright holder must demonstrate or obtain permission to publish this material under the Creative Commons Attribution 4.0 International (CC BY 4.0) License used by PLOS journals. Please closely review the details of PLOS’s copyright requirements here: PLOS Licenses and Copyright. If you need to request permissions from a copyright holder, you may use PLOS's Copyright Content Permission form.

Potential Copyright Issues:

Figure 2: please (a) provide a direct link to the base layer of the map (i.e., the country or region border shape) and ensure this is also included in the figure legend; and (b) provide a link to the terms of use / license information for the base layer image or shapefile. We cannot publish proprietary or copyrighted maps (e.g. Google Maps, Mapquest) and the terms of use for your map base layer must be compatible with our CC-BY 4.0 license. 

Additional Editor Comments (if provided):

Reviewers' comments:

Reviewer's Responses to Questions

**Comments to the Author**

1. Does this manuscript meet PLOS Global Public Health’s publication criteria? Is the manuscript technically sound, and do the data support the conclusions? The manuscript must describe methodologically and ethically rigorous research with conclusions that are appropriately drawn based on the data presented.

Reviewer #1: Yes

Reviewer #2: Partly

Reviewer #3: Partly

2. Has the statistical analysis been performed appropriately and rigorously?

Reviewer #1: I don't know

Reviewer #2: I don't know

Reviewer #3: I don't know

3. Have the authors made all data underlying the findings in their manuscript fully available (please refer to the Data Availability Statement at the start of the manuscript PDF file)?

Reviewer #1: Yes

Reviewer #2: Yes

Reviewer #3: Yes

4. Is the manuscript presented in an intelligible fashion and written in standard English?

Reviewer #1: Yes

Reviewer #2: Yes

Reviewer #3: Yes

5. Review Comments to the Author

Reviewer #1: This is the first time I reviewed this manuscript. I thought it was excellent. It's unique and well written. They are clear about their obvious limitations. I am not familiar with these methods but they appear sounds.

Reviewer #2: •  It reads more like a demonstration of a technique rather than a substantial contribution to the FGM debate. It is a neat demonstration, but ' so what'.

• On p. 5, 2nd paragraph the authors state what they will look at but no argument is made why exactly these aspects of the Twitter conversations are studied or what their relevance is.

• A demonstration of a new technique can be valuable but then the authors need to provide more details about how they went about it and what choices were made for what reasons. This is all lacking in this paper where the techniques and choices are only described in very general terms. Such demonstrations are only useful when they ' teach' how to use the technique.

• For instance, I would like to know how the tweets studied were selected. In meta-studies, for instance, one needs to specify in detail how the studies were selected. A similar approach would be useful here.

• In figure 1 the authors list the most frequent terms in 2020. Although the figure is unreadable in the manuscript file one should provide a list of the most frequent word for each of the years, and also say something about how this changes over time and how this should be interpreted.

• Likewise, on p. 6 the authors state that 34 top words in the text were distinguished. Again I would like to know what these words were and how their use changes over time.

• p.6: Explain how 10 topics were selected after ' exploratory analysis' .

• I do have my doubts limiting the analysis to English language tweets. It shouldn't come as a surprise that most tweets come from the US and the UK and some other large English-language countries. However, these usually are not the countries where FGM is most prevalence. In both the US and the UK the practice is banned and rare, while although both Nigeria and Kenya do know FGM the prevalence there is only 27%. Therefore, the countries where the practice is most common are excluded from the analysis because of the choice to include only tweets in English. I also wonder what the relevance of the US and UK tweets is for the struggle against FGM. This also introduces a serious bias in the data.

• On p.8 the authors report the number of users over the various year. A substantial variation is observed with especially in 2016 a huge drop in the number of users. I would like to read some tentative explanation for these fluctuations.

• I am surprised by the observation that the description terms do not vary substantially over time, while I would expect that terms like ' Trump' and ' MAGA' to be still relatively rare in 2015.

• What do the shifts in topics described in Table 2 mean. Please provide some tentative interpretation.

Reviewer #3: The manuscript describes tweets about female genital mutilation/cutting in English from the years 2015-2020, using machine learning methods. The aim is to characterize the substance of conversations about FGM/C during these six years on Twitter and assess changes over time. The topic of public discourse of FGM/C is an interesting topic that would be of interest to many readers and I applaud the researchers for initiating this research, although I questions whether PLOS Global Public Health is the most appropriate journal. As a reviewer asked to critically examine the manuscript, I have concerns about the authors' presentation of the study, the rigor of the research methods, and hence the validity of the findings. I offer some comments below, which may serve the authors in their future work.

Clearly, I find that the methods are the most worrisome part of the study presented. First, the aim of the study is expressed in vague terms (page 5: “to characterize the substance of conversations about FGM/C on Twitter between 2015-2020 and assess any changes over time”). It is not stated how the tweets were recruited/extracted/retrieved from Twitter. It appears that the unit of analysis was a tweet addressing FGM/C (indicated by e.g. the title and the discussion stating “Tweets about FGM/C were identified based on the presence of certain terms” page 15), but it could also be Twitter user (who tweeted about FGM/C). In any case, the researchers do not state what the inclusion criteria were. They need to explain each variable and how it was defined. On page 6 they explain that they evaluated the use of FGM/C terminology by determining the frequencies of the terms, but clearly, this must be closely link to their inclusion criteria leading to a tweet (or user) being included in the first place so this must be addressed. On page 7 they state that tweet topics were synthesized, but they do not explain how. It appears that the machine learning methods that the researchers used were two classifiers, and they were only used to classify user engagement, but this is not clear and the two classifiers are not described. On page 12, the researchers state that random forest model was used to predict engagement and they should state what the predictors in the model were and how engagement was defined. In sum, my main concern is the careless (unscientific) description of the research conducted.

In the results section, I suggest the researchers follow standard reporting and start by presenting the sample. The “full dataset” is referred to on page 12 (line 239) and it is unclear if the full dataset is the 338,845 unique users they refer to on page 8 or the “large sample size … With over a million original tweets” (on page 15). They explain that the number of verified Twitter users were stable, but what does it mean in this study that a user is verified? If it means something along the lines of having the blue checkmark (to indicate active, notable, and authentic accounts that Twitter verified based on certain requirements) then it must be stated and the discussion needs to address that >90% of users in this study were NOT verified. Related, in the introduction the researchers state that “social media are uniquely positioned to track public conversations over time, particularly for sensitive topics” but what is the evidence that we can trust social media posts (tweets in this care) about a topic are expressions of people’s true sentiments? In the results, the researchers should re-consider analysing based on country, given that only a third of the users (or tweets) could be linked to a country (i.e. 70% of the sample missing). Without a description of the inclusion criteria it is impossible to judge the results regarding the terms used to refer to FGM/C, but presumably to be included in the dataset certain terms used for FGM/C had to be used. It is a stretch to claim that a term used 2.2% of the time is a “commonly used” term (page 19). In table 2 the topics listed in column 1 are presumably labelled by the authors and I suggest they re-consider the labels. Judged by the tweet examples in the table, it is hard to see the difference between some of the topics, e.g. the topic ‘cultural aspects’ and ‘activism. ‘Evils of the other’ could more appropriately be labelled violence against women. It is very hard to believe that the topics of ‘medical risks’ and ‘cultural aspects’ were not topics at all in four and three years, respectively. Findings that the highest volume of FGM/C related tweets was on February 6 is obvious.

The discussion should include less repetition of the results and more what the results mean, what are their implications, without being too speculative. The researchers should be careful drawing conclusions. First, at all, because as they state themselves “Twitter users are not representative of the global public” and second, about users’ country of residence, given they only included English language tweets. Clearly, when the absolute number of Twitter users increase, one can expect to observe also an increase in tweets addressing FGM/C. On page 14 (lines 275-276) it is incorrect to state that feminism and human rights were among the persistent topics – they are at the bottom of the table (table 2) and only identified two of the six years. Other topics are identified all or most years. In the discussion, they need to take into account the change in number of Twitter users/accounts during the years analysed. The authors give five implications of their findings, but they do not seem substantive. Additionally, the first appears tautological (linked to inclusion of tweets), the third is dubious because only 7% of users were verified, the fourth is redundant (those of us who have researched and been engaged with the topic of FGM/C know that FGM/C is politicized) and the fifth is not linked to any finding in this study and is normative.

A few other, minor concerns are: It is not clear what they mean when they state that: the findings can be used to “leverage social media” (page 5), that the analysis is the “first of its scale” (page 5, what do they mean by ‘scale’ in this context?), “we evaluated changes in the most frequently identified countries across years” (page 6, what do they mean by changes in countries?), “phenomena that are considered ‘of the other’ in the West” (page 14). There are also several grammatical and spelling mistakes that should be corrected.

6. PLOS authors have the option to publish the peer review history of their article (what does this mean?). If published, this will include your full peer review and any attached files.

**Do you want your identity to be public for this peer review?** For information about this choice, including consent withdrawal, please see our Privacy Policy.

Reviewer #1: **Yes: **Nicole Warren

Reviewer #2: No

Reviewer #3: No

---

## [Decision Letter · Decision Letter 1]

7 Jun 2023

Use of machine learning methods to understand discussions of female genital mutilation/cutting on social media

PGPH-D-22-01078R1

Dear Ms Weber,

We are pleased to inform you that your manuscript 'Use of machine learning methods to understand discussions of female genital mutilation/cutting on social media' has been provisionally accepted for publication in PLOS Global Public Health.

Best regards,

Julia Robinson

Executive Editor

Reviewer Comments (if any, and for reference):

Reviewer's Responses to Questions

**Comments to the Author**

1. If the authors have adequately addressed your comments raised in a previous round of review and you feel that this manuscript is now acceptable for publication, you may indicate that here to bypass the “Comments to the Author” section, enter your conflict of interest statement in the “Confidential to Editor” section, and submit your "Accept" recommendation.

Reviewer #4: All comments have been addressed

2. Does this manuscript meet PLOS Global Public Health’s publication criteria? Is the manuscript technically sound, and do the data support the conclusions? The manuscript must describe methodologically and ethically rigorous research with conclusions that are appropriately drawn based on the data presented.

Reviewer #4: Yes

3. Has the statistical analysis been performed appropriately and rigorously?

Reviewer #4: N/A

4. Have the authors made all data underlying the findings in their manuscript fully available (please refer to the Data Availability Statement at the start of the manuscript PDF file)?

Reviewer #4: Yes

5. Is the manuscript presented in an intelligible fashion and written in standard English?

Reviewer #4: Yes

6. Review Comments to the Author

Reviewer #4: (No Response)

7. PLOS authors have the option to publish the peer review history of their article (what does this mean?). If published, this will include your full peer review and any attached files.

**Do you want your identity to be public for this peer review?** For information about this choice, including consent withdrawal, please see our Privacy Policy.

Reviewer #4: No
